# Sex-specific triacylglycerides are widely conserved in *Drosophila* and mediate mating behavior

**Jacqueline SR Chin[1,2], Shane R Ellis[3†], Huong T Pham[3‡], Stephen J Blanksby[3], Kenji Mori[4], Qi Ling Koh[1], William J Etges[5], Joanne Y Yew[1,2]\***

[1]Temasek Life Sciences Laboratory, National University of Singapore, Singapore, Singapore; [2]Department of Biological Sciences, National University of Singapore, Singapore, Singapore; [3]ARC Centre of Excellence for Free Radical Chemistry and Biotechnology, University of Wollongong, Wollongong, Australia; [4]Photosensitive Materials Research Center, Toyo Gosei Company, Ltd., Chiba, Japan; [5]Program in Ecology and Evolutionary Biology, Department of Biological Sciences, University of Arkansas, Fayetteville, United States

**Abstract** Pheromones play an important role in the behavior, ecology, and evolution of many organisms. The structure of many insect pheromones typically consists of a hydrocarbon backbone, occasionally modified with various functional oxygen groups. Here we show that sex-specific triacylclyerides (TAGs) are broadly conserved across the subgenus *Drosophila* in 11 species and represent a novel class of pheromones that has been largely overlooked. In desert-adapted drosophilids, 13 different TAGs are secreted exclusively by males from the ejaculatory bulb, transferred to females during mating, and function synergistically to inhibit courtship from other males. Sex-specific TAGs are comprised of at least one short branched tiglic acid and a long linear fatty acyl component, an unusual structural motif that has not been reported before in other natural products. The diversification of chemical cues used by desert-adapted *Drosophila* as pheromones may be related to their specialized diet of fermenting cacti.

**\*For correspondence:** joanne@tll.org.sg

**Present address:** [†]Biomolecular Imaging Mass Spectrometry, FOM Institute AMOLF, Amsterdam, Netherlands; [‡]Department of Chemistry, University of California, Riverside, Riverside, United States

**Competing interests:** The authors declare that no competing interests exist.

## Introduction

Chemical communication significantly influences many complex social behaviors, including aggression, kin recognition, and courtship (**Wyatt, 2003**). The chemical structures and functions of insect pheromones have been intensely studied because of the fascinating diversity of behavioral properties and relevance to questions of speciation, reproductive isolation, and applications to pest control (**Witzgall et al., 2010**). Since the discovery of Bombykol in 1959 (**Butenandt et al., 1959**), hundreds of insect pheromones have been identified, including straight chain and branched alkanes and alkenes, oxygen-containing derivatives such as wax esters, fatty alcohols, and aldehydes, sterols, and isoprene-based compounds (**Tillman et al., 1999**; **El-Sayed, 2012**).

In *Drosophila*, pheromones are produced by oenocytes (specialized epithelial cells in both males and females) and the male ejaculatory bulb and subsequently secreted onto the cuticular surface and anogenital region, respectively (**Billeter et al., 2009**; **Yew et al., 2009**). Previous studies of *Sophophora* and *Drosophila* flies identified alkanes, alkenes, and oxygen-modified hydrocarbons as the major lipids used as pheromones (**Jallon and David, 1987**; **Greenspan and Ferveur, 2000**). Recently, triacylglycerides (TAGs), which are normally found in the fat bodies and used for energy storage, were observed on the cuticles of flies from the *Drosophila repleta* and *Drosophila quinaria* groups (**Yew et al., 2011**; **Curtis et al., 2013**). However, almost nothing is known about the

**eLife digest** For animals, the ultimate purpose of life is to have sex, as nothing is more important than passing down your genes to future generations. A wide range of strategies are therefore employed throughout nature to maximize the chances of sexual success, from ostentatious courtship rituals to the subtle subliminal signals sent out using chemicals called pheromones. Plants and animals release pheromones to influence the behavior of other plants and animals, often without the recipient being aware of it.

Hundreds of different insect pheromones have been discovered. Fruit flies release a number of different pheromones, all with similar chemical structures. Now, Chin et al. have discovered that male flies belonging to several species of fruit fly that live in the desert release chemicals called triacylglycerides (TAGs), which are commonly used for energy storage by many organisms as pheromones. During sex, the male fly rubs the TAGs onto the body of the female, which makes her less attractive to other male flies for several hours, thus increasing his chances of parenthood and passing his genes to future generations.

TAGs are also found in other insect species, but have been largely overlooked as pheromones. Moreover, the TAGs discovered by Chin et al. have an unusual structure, not previously seen in nature, which may result from the diet of fermenting cacti the desert-dwelling fruit flies enjoy.

structure, chemical diversity, conserved expression, and functional roles of these exogenously secreted TAGs.

To explore the role of TAGs as pheromones and the ubiquity of their expression in *Drosophila*, we used ultraviolet laser desorption/ionization mass spectrometry (UV-LDI MS) to analyze the cuticular profiles of flies from seven major *Sophophora* and *Drosophila* groups. We also investigated the chemical structures of sex-specific TAGs and their role as sex pheromones in species from the *D. repleta* group. Our studies indicate that TAGs are a broadly conserved, structurally atypical class of *Drosophila* pheromones that has been overlooked.

## Results

### Sex-specific triacylglycerides are conserved in other drosophilids

We used UV-LDI MS to perform a broad survey of cuticular lipid profiles of flies from the *Drosophila* and *Sophophora* subgenera. UV-LDI MS provides spatially resolved chemical profiling from single, intact insects by probing the cuticular surface with a 200 μm laser (*Yew et al., 2009*, *2011*). Chemical signatures consistent with TAG structures were found to be largely conserved across 3 different *Drosophila* groups: the repleta radiation (including *Drosophila hydei*, *Drosophila buzzatii*, *Drosophila navojoa*, *Drosophila wheeleri*, and *Drosophila aldrichi*), the virilis group (*Drosophila americana*, *Drosophila virilis*, and *Drosophila montana*), and within the robusta group (*Drosophila robusta*) (*Figure 1*; *Figure 1—figure supplement 1*). The TAGs were expressed only in the ejaculatory bulb of males. In contrast, sex-specific TAGs were not detected in any of the species tested from the *Sophophora* subgenus. Many of the TAG-producing species are capable of feeding and reproducing on cacti, fungi (mushroom), and tree sap or slime fluxes, substrates that contain high levels of toxins, plant defensive compounds, or bacteria, a characteristic that may be related to their ability to produce sex-specific TAGs.

### Sex-specific TAGs are correlated with age and synthesized in the ejaculatory bulb

To characterize the structures and functions of sex-specific TAGs, we focused on desert-adapted drosophilids from the *D. repleta* group, *Drosophila arizonae* and *Drosophila mojavensis*, two well-characterized models for speciation, reproductive isolation, and ecological studies (*Ruiz et al., 1990*; *Markow, 1996*; *Etges and Jackson, 2001*). Analysis by UV-LDI MS detected 13 TAGs and several long-chain acetyldienyl acetates, 30 or 32 carbons in length (referred to as long OAcs) exclusively in the anogenital region of *D. arizonae* and *D. mojavensis* males and not on virgin females of either species (*Figure 2A,B*; *Figure 2—figure supplement 1*).

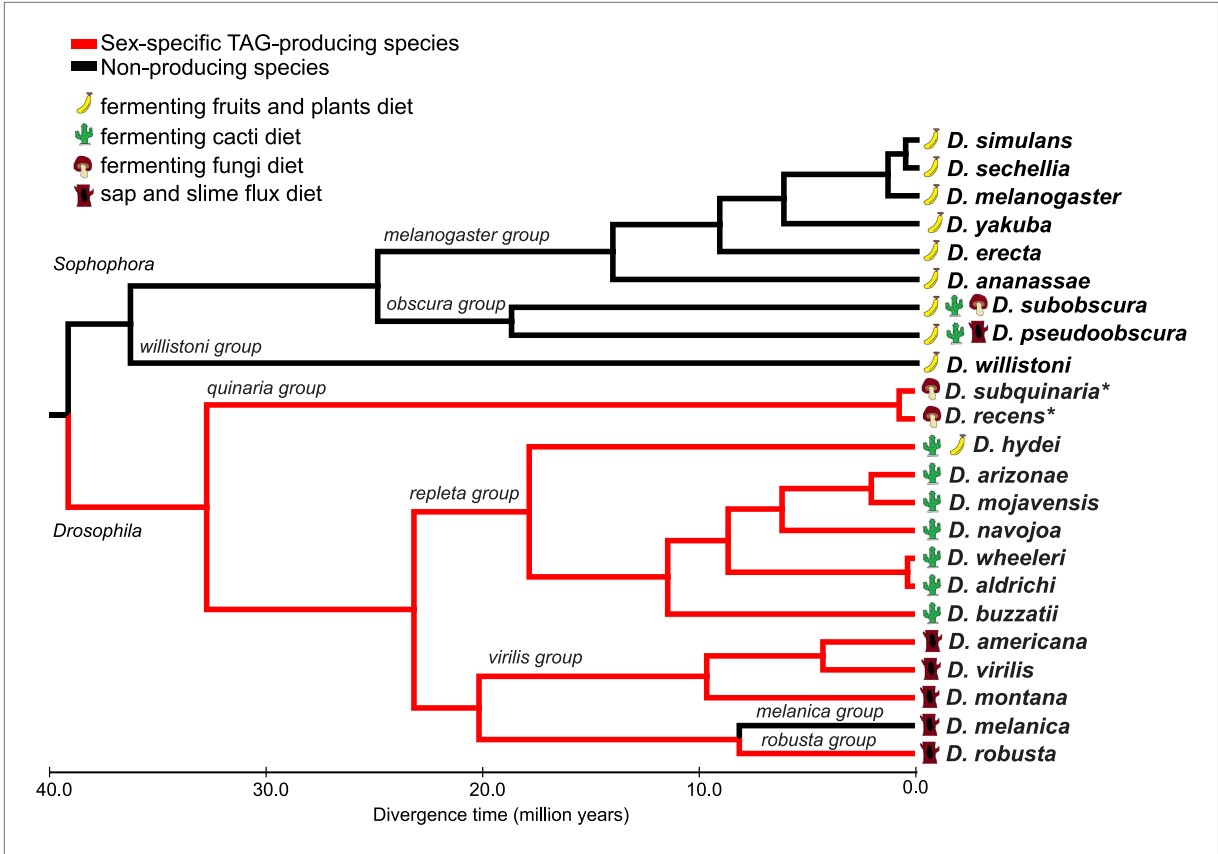

**Figure 1**. Male-specific TAG expression is broadly conserved across the *Drosophila* subgenus and not found in species from *Sophophora*. The primary diets of each species are indicated, based on the previous studies. Branches for TAG-producing species are colored in red. Branch lengths are proportional to evolutionary time. *Evidence for TAG-expression is based on *Curtis et al., 2013*.

The following figure supplements are available for figure 1:

**Figure supplement 1**. Representative UV-LDI spectra from distantly related drosophilids in the *Drosophila* subgenus.

We next tested whether sex-specific TAG expression is correlated with male sexual maturity. Chemical profiling of *D. arizonae* from 0 to 15 days old indicated that the expression of the TAGs and long OAcs increased in abundance as males get matured, with little or no expression in the first 4 days after eclosion and higher expression towards the age of maturity at approximately 8 day old, the age when males exhibit full courtship behaviors (*Markow, 1981*) (*Figure 2C*). *D. mojavensis* followed a similar maturation profile (*Figure 2D*). Analysis of dissected male reproductive organs by UV-LDI MS revealed qualitatively similar chemical profiles exclusively in the ejaculatory bulbs (*Figure 2—figure supplement 1*). In addition, no predicted precursors of these compounds such as diacylglycerol or glycerol-3-phosphate were detected from the accessory glands or other reproductive organs. These results indicate that the TAGs and long OAcs are synthesized in the ejaculatory bulb.

## Sex-specific TAGs exhibit unusual structural features and are expressed as a complex blend of isomers

From *D. arizonae* and *D. mojavensis*, we isolated both TAG and long OAc lipid classes using thin layer chromatography (TLC; *Figure 3—figure supplement 1*). Chemical derivatization of the long OAc fraction confirmed the presence of an acetyl group (*Figure 3—figure supplement 2*). Gas chromatography MS (GCMS) analysis of transesterified TLC fractions indicated tiglic acid, a 5-carbon branched unsaturated acid, as one of the fatty acyl moieties (*Figure 3—figure supplement 3*). No other 5-carbon fatty acid methyl esters were detected. Tandem MS with low energy collision-induced dissociation (CID) analysis provided the chain length and degree of unsaturation of each of the acyl chains present

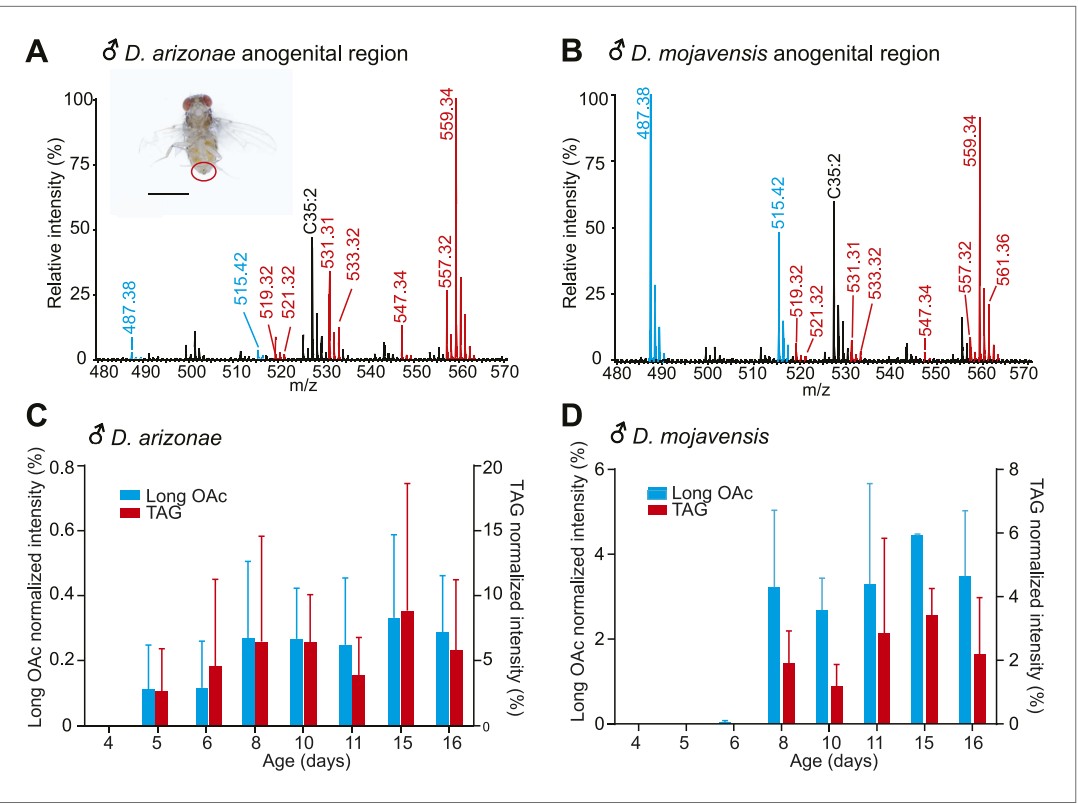

**Figure 2**. Pheromone profiles and age-related increase in sex-specific TAGs. (**A** and **B**) UV-LDI MS allows spatially resolved detection of high molecular weight lipids directly from intact insects, with minimal damage to the cuticle. Representative mass spectra from the anogenital region (inset) of *D. arizonae* and *D. mojavensis* males show signals corresponding to triglycerides (TAGs, red) and long chain alkadienyl acetates (long OAcs, blue). The hydrocarbon C35:2 (number of carbons: number of double bonds) is found on cuticles of males and females. Labeled signals correspond to potassiated molecules [M + K]+. Scale bar: 1 mm. (**C** and **D**) Relative intensity of TAGs and long OAcs on male *D. arizonae* and *D. mojavensis*, respectively. TAGs and long OAcs increase with age, with trace quantities first appearing at 4 day old. The signal intensity for all detected TAGs or long OAcs was normalized to the signal intensity of C35:2.

The following figure supplements are available for figure 2:

**Figure supplement 1**. UV-LDI MS profiles from the anogenital region of virgin females and dissected ejaculatory bulb (eb) and accessory glands.

in each TAG. A similar motif was revealed among each of the molecules: a single long-chain fatty acyl component, 16–18 carbons in length, together with 2 short-chain fatty acyl side chains, each 2–5 carbons in length (*Figure 3A*; *Figure 3—figure supplement 4*). For several of the more abundant TAG molecules, the position of the acyl chains on the glycerol backbone could be deduced based on the relative abundance of the product ions in the CID spectra. As described by *Hsu and Turk (1999 and 2010)*, fragments reflecting the loss of substituent at the *sn*-2 carbon (middle of the glycerol backbone) are less abundant than ions reflecting losses at either the *sn*-1 or *sn*-3 carbons. Based on this observation, long-chain fatty acids are predominantly located at either *sn*-1 or *sn*-3 for the major TAGs at [M + Na]+ 543 and 541 (*Figure 3A*; *Figure 3—figure supplement 4*) and [M + Li]+ 499, 501, and 527 (*Figure 3—figure supplement 4*). It was not possible to distinguish between *sn*-1 and *sn*-3 positions. The position assignments on the backbone are supported by analysis of synthetic standards in which a long-chain fatty acid is placed at *sn*-1. The relative abundances of fragment ions are similar to those observed from crude extract (*Figure 3—figure supplement 6–8*). Notably, CID analysis of a fourth major TAG at [M + Li]+ 487 resulted in low-intensity signals corresponding to loss of the C18:1 fatty acyl substituent, suggesting that this component is likely to reside at the

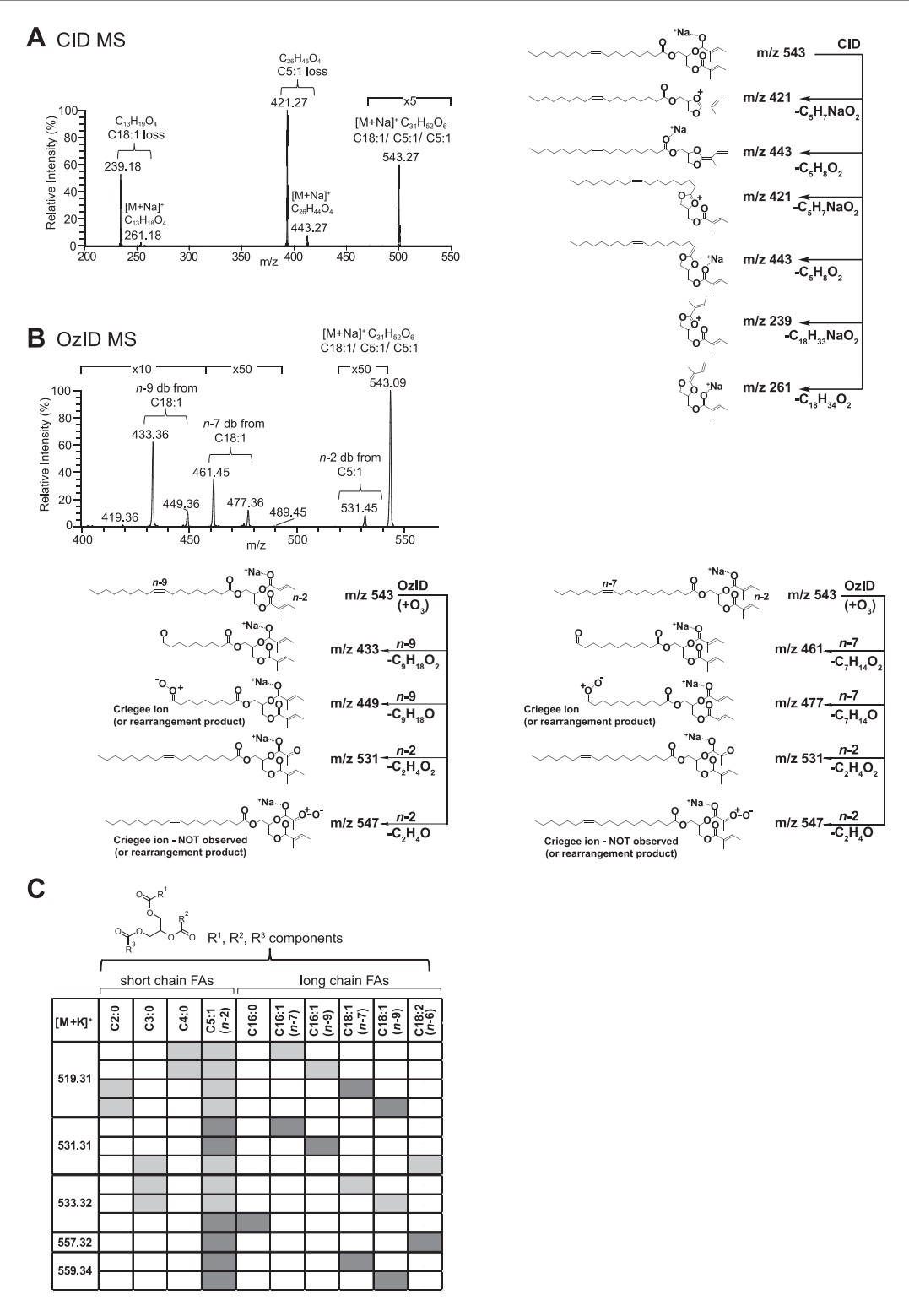

**Figure 3**. Structural elucidation of sex-specific TAGs. (**A**) The low energy collision-induced dissociation (CID) mass spectrum of a TAG-related signal from crude *D. arizonae* extract ([M + Na]⁺ 543) shows fragments corresponding to losses of a 5 carbon fatty acid with a single double bond (C5:1) and an 18 carbon fatty acid with a single double bond (C18:1). Both sodiated (major peak) and protonated chain side losses are observed. The schematic rationalizes the product ions formed during CID of mass-selected [M + Na]⁺ of unsaturated lipids. (**B**) Ozone-induced
*Figure 3. Continued on next page*

*Figure 3. Continued*

dissociation (OzID) of a TAG-related signal (shown in **A**) indicates isomers with variant double bond positions. The fragments at *m/z* 461 and *m/z* 433 are aldehyde products consistent with double bonds (db) at positions *n*-7 and *n*-9, respectively. The fragment at *m/z* 531 confirms the *n*-2 double bond position found in the tiglic acyl component. The corresponding Criegee product ions (*m/z* 477 and *m/z* 449, respectively) are also observed. The schematic rationalizes the product ions formed during OzID of mass-selected [M + Na]⁺ of unsaturated lipids. Product ions are assigned as outlined by ***Thomas et al., 2008*** and ***Brown et al., 2011***. (**C**) CID and OzID MS analyses of the most abundant sex-specific TAGs reveal significant combinatorial complexity. A generic TAG molecule consisting of a glycerol backbone and 3 fatty acyl (FA) side chains, R1, R2, and R3, is shown. Each TAG species is comprised of 2 short chain and 1 long chain FA component. Shaded boxes indicate the composite side chains of each TAG species. The glycerol backbone positions for several TAGs are assigned based on the comparison with synthetic standards and ion product abundance patterns (dark gray boxes). Ambiguous backbone positions are in light gray.

The following figure supplements are available for figure 3:

**Figure supplement 1**. Thin layer chromatography (TLC) separation of *D. arizonae* male cuticular lipid extract.

**Figure supplement 2**. Direct analysis in real time (DART) MS spectrum of the TLC fraction containing long OAcs after derivatization by base hydrolysis confirms the presence of an acetyl functional group.

**Figure supplement 3**. Structural elucidation of sex-specific TAGs using gas chromatography MS.

**Figure supplement 4**. Structural elucidation of TAGs by CID MS reveals fatty acid (FA) components with 2, 3, 5, 16, or 18 carbons in length and 0–2 double bonds.

**Figure supplement 5**. Analysis of TAGs by OzID reveals double bond positions of acyl side chains.

**Figure supplement 6**. Spectra obtained from CID MS and OzID analyses of a synthetic TAG comprised of an oleic acid (*cis*-9-Octadecenoic acid) and tiglic acid side chains are consistent with the analysis of a TAG molecule with identical *m/z* found from crude extract.

**Figure supplement 7**. The spectrum obtained from CID MS analysis of a synthetic TAG (16:1/5:1/5:1) is consistent with the spectrum from a TAG molecule with identical *m/z* found from crude extract.

**Figure supplement 8**. Spectra obtained from CID MS and OzID analyses of a synthetic TAG consisting of linoleic acid (*cis, cis*-9,12-Octadecadienoic acid) and tiglic acid side chains are consistent with spectra from analysis of a TAG molecule with identical *m/z* found from crude extract.

*sn*-2 position (***Figure 3—figure supplement 4***). Low-abundance signals for isobaric TAGs containing C18:2 and C18:1 fatty acids were also observed at [M + Li]⁺ 501, 499, and 487 and are likely to represent minor components. In these cases, it was not possible to assign substituent positions.

To determine the double bond positions within each fatty acid, we used ozone-induced dissociation (OzID) mass spectrometry (***Thomas et al., 2008***; ***Brown et al., 2011***). Individual TAG species were mass-selected within an ion-trap mass spectrometer where they were exposed to ozone vapor. The resulting gas-phase ion–molecule reaction facilitates targeted oxidative dissociation of carbon–carbon double bonds present in the acyl chains. Fragmentation of the ozonide leads to formation of characteristic aldehyde and Criegee ions with a mass indicative of the positions of each double bond. OzID analysis of the TAG fraction revealed numerous positional isomers, with double bond positions between C9-C10 (*n*-9) and C11-C12 (*n*-7), indicating oleic and palmitoleic acid side chains, and between C2-C3 (*n*-2), consistent with tiglic acid (***Figure 3B,C***; ***Figure 3—figure supplement 5***). Spectra from OzID analysis of TAG standards synthesized with oleic acid (C18:1, *n*-9) or linoleic acid (C18:2, *n*-6,9) support the double bond position assignments (***Figure 3—figure supplement 6 and 8***). Double-bond geometry could also be deduced for two of the more abundant TAGs. *cis*- and *trans*-alkenes exhibit differential reactivity to ozone, resulting in differences in the overall abundances of the fragment ions and the relative abundance of the Criegee and aldehyde product ions (***Poad et al.,***

*2010*). The relative abundance of the aldehyde and Criegee ions for the molecules at $[M + Na]^+$ 543 and 541 are consistent with those of synthetic TAG standards synthesized with oleic acid and linoleic acid, both of which contain *cis*- double bonds (*Figure 3—figure supplement 6 and 8*). In summary, MS analysis revealed considerable variation in the carbon chain length, degree of unsaturation, positions of fatty acyl chains, and double bond positions of both the short chain and long chain fatty acyl components (*Figure 3C*). All of the analyzed TAGs contained tiglic acid. The unusual combination of short odd-branched chain fatty acids with a single linear long-chain component has not been reported before in natural products.

### Diet-related effects on sex-specific TAGs

To determine the contribution of diet to TAG production, we compared TAG levels between males raised on standard fly media for 2 generations vs media supplemented with cactus powder and banana. Thin layer chromatography of the lipid contents of ejaculatory bulbs indicated that males raised on standard media produced a significantly lower amount of some of the sex-specific TAGs, including $[M + K]^+$ 559, one of the most abundant molecules (*Figure 4*). The results show that although a specialized diet is not essential for sex-specific TAG production, precursors derived from food can influence the quantity of several of the TAGs.

### Sex-specific lipids are transferred to females during mating and are correlated with loss of female attractiveness

In some species of insects, males anoint the females with anti-aphrodisiacs during mating to suppress subsequent courtship from other males (*Zawistowski and Richmond, 1986*; *Bownes and Partridge, 1987*; *Wigby et al., 2009*; *Yew et al., 2009*). We hypothesized that sex-specific TAGs may play a similar role based on the sexually dimorphic pattern of expression and localization to a male sex organ. To test this prediction, a mate choice assay was used in which a naïve male was given a choice to court either a virgin female or a recently mated female (*Figure 5A*). Male *Drosophila* courtship behavior consists of a sequence of stereotyped, quantifiable features, including wing vibration ('singing'), foreleg tapping, proboscis extension, and copulation (*Spieth, 1974*). Courtship initiation and copulation preferences were measured since both indicate male choice while the latter is also influenced by female rejection behavior. Males from *D. arizonae* and *D. mojavensis* were significantly more attracted to virgin females than recently mated females (*Figure 5B*). Notably, significant levels of both TAGs and long OAcs were found on the anogenital regions of *D. arizonae* females shortly after mating but decreased by approximately 80% at 2–4 hr post-mating and were almost negligible at 8 hr post-mating (*Figure 5C*; *Figure 5—figure supplement 1*). Mated *D. arizonae* females became increasingly attractive over time, correlating with a decrease of the levels of TAGs and long OAcs on the cuticle (*Figure 5D*). From 4 hr onwards, males showed no significant preference between mated and virgin

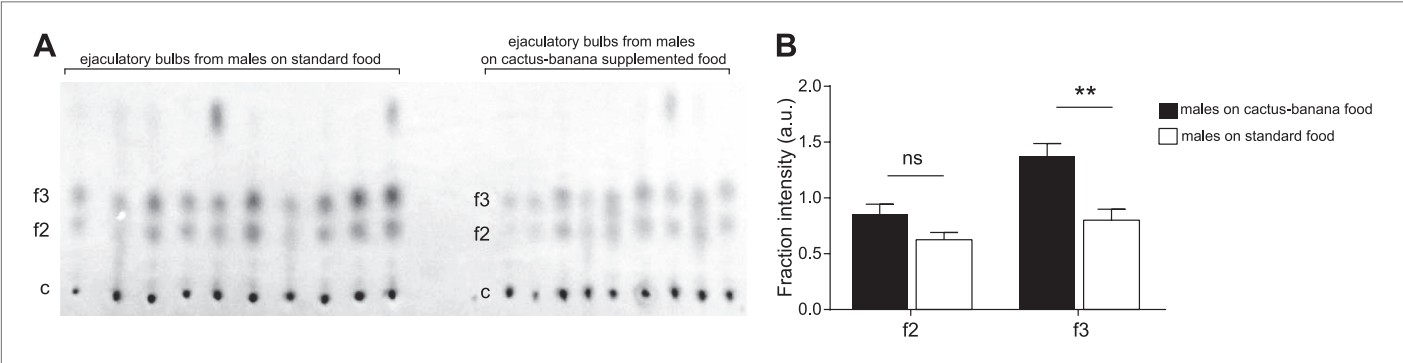

**Figure 4**. Diet changes the quantity but not composition of sex-specific TAGs. (**A**) TAGs from individual ejaculatory bulbs of males raised on standard fly food (n = 10) or cactus-banana supplemented food (n = 9) were quantified using direct tissue thin layer chromatography. Each lane contains a single bulb. c: control band (point of origin) used for normalization; f2 and f3: fractions containing sex-specific TAGs. (**B**) The amount of TAGs in f3 from ejaculatory bulbs of males raised on standard food is significantly lower than supplemented food conditions (Student's t-test, two-tailed, p=0.0016). TAGs found in f2 were not significantly different (p=0.062). Error bars indicate SEM. **\*\***: p<0.005; ns: not significant; a.u.: arbitrary units.

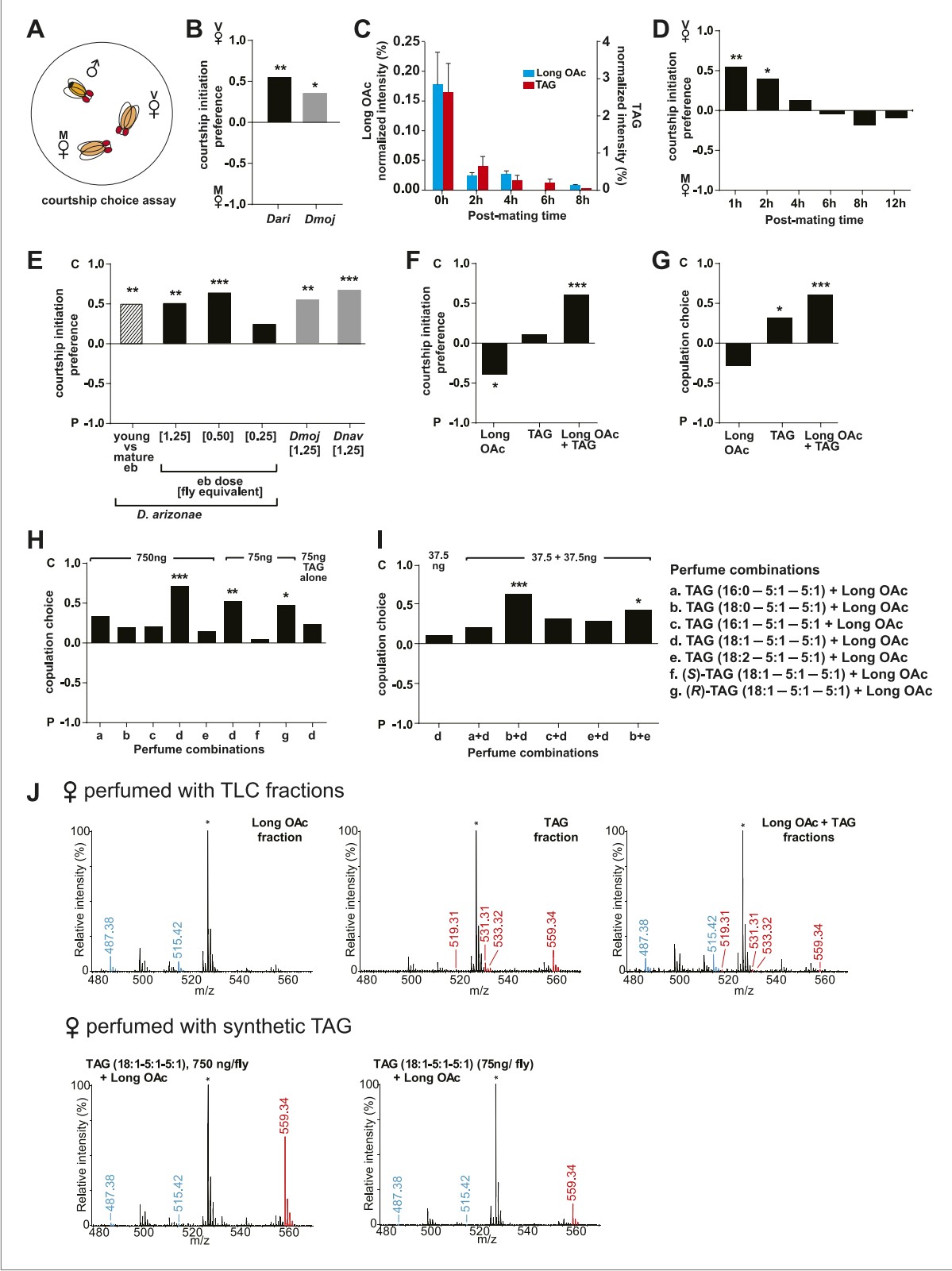

**Figure 5**. Sex-specific lipids suppress male mating behavior. (**A**) To measure male courtship behavior, one male fly is placed with 2 females, one mated (M), and one virgin (V). (**B**) *D. arizonae* (*Dari*) and *D. mojavensis* (*Dmoj*) prefer to court virgin females over recently mated females (n = 20, Fisher's exact test, p=0.00123; n = 31, p=0.0105). **: p<0.01; ns: not significant. A preference score of 1 indicates all males initiate courtship first with the virgin female; −1 indicates all males initiated courtship first with the mated female. (**C**) Levels of male-transferred TAGs and long OAcs on the female cuticle

*Figure 5. Continued on next page*

*Figure 5. Continued*

after first mating decreases by 2 hr post-mating. (**D**) Females are significantly less attractive for up to 2 hr after mating. By 4 hr, males do not exhibit significant preference for courting mated vs virgin females. **: Fisher's exact test, p=0.00123; *: p=0.0256. (**E**) *D. arizonae* males are more reluctant to initiate courtship with females perfumed with the contents of [1.25] ejaculatory bulb (eb) (n = 27, Fisher's exact test, p=0.000624) or [0.5] eb (n = 28, p<0.0001) but not [0.25] eb (n = 28, p=0.176). Extracts from immature male ebs were ineffective at inhibiting male courtship (n = 28, p=0.000389). *D. mojavensis* and *D. navojoa* (*Dnav*) males also avoided virgin females perfumed with eb contents (*Dmoj*: n = 21, p=0.00160; *Dnav*: n = 21, p<0.0002). **: p<0.002; ***: p<0.0001. C: solvent control; P: perfumed. (**F**) Suppression of *D. arizonae* courtship initiation is elicited only when TAGs and long OAcs are combined (n = 28, Fisher's exact test, p<0.0001). TAGs alone are ineffective (n = 28, p=0.593). Long OAcs on their own could be attractive to males (n = 28, p=0.006). *: p<0.05; ***: p<0.0001. (**G**) *D. arizonae* copulation is suppressed in the presence of TAGs alone (n = 28; Fisher's exact probability test, p=0.0287) or TAGs combined with long OAcs (n = 28; p<0.0001), but not long OAcs alone (n = 28; p=0.0543). *: p<0.05; ***: p<0.0001. A copulation choice score of 1 indicates all males copulated with solvent-perfumed females; −1 indicates all males copulated with TAG-perfumed females. (**H**) Perfuming with synthetic TAGs recapitulates copulation suppression. Oleic acid (C18:1)-containing TAGs produced significant effects at high and low doses (750 ng: n = 21, Fisher's exact test, p<0.0001; 75 ng: n = 21, p=0.00167). Only the (*R*)-18:1 stereoisomer was bioactive (75 ng: n = 21, p=0.00480); the (*S*)-18:1 stereoisomer did not elicit a significant behavioral response (75 ng: n = 21, p=1). *: p<0.01; **: p<0.002; ***: p<0.0001. (**I**) Two combinations of TAGs produced synergistic effects on copulation suppression: oleic acid-TAG paired with stearic acid-TAG (n = 21, p=0.000139) and stearic acid-TAG paired with linoleic acid-TAG (n = 19, p=0.022). The oleic acid-containing TAG is not bioactive at a dose of 37.5 ng/fly (n = 20, Fisher's exact test, p=0.751). *: p<0.05; ***: p<0.0005. (**J**) UV-LDI MS spectra of females perfumed with TLC fractions or synthesized TAGs. Signals for TAGs and long OAcs are indicated in red and blue, respectively. *: C35:2 Pentatriacontadiene reference peak, m/z 527.5 $[M + K]^+$.

The following figure supplements are available for figure 5:

**Figure supplement 1**. UV-LDI spectra of mated *D. arizonae* female anogenital regions reveal signals corresponding to male-specific TAGs (red) and long OAcs (blue) for up to 12 hr post-mating.

**Figure supplement 2**. *D. arizonae* mated females are receptive to copulation starting at 8 hr after the first mating.

**Figure supplement 3**. *D. arizonae* males avoided copulating with females perfumed with the contents of male ejaculatory bulbs (eb) at a concentration of [1.25] eb (n = 27, Fisher's exact test, p=0.0003) or [0.5] eb (n = 28, Fisher's exact test, p<0.0001) but not [0.25] eb (n = 28, p=0.176).

**Figure supplement 4**. *D. arizonae* courtship and copulation preferences towards partly olfactory and gustatory perception-deficient virgin females that are perfumed with control or ejaculatory bulb extract.

**Figure supplement 5**. *D. arizonae* courtship preferences towards virgin females perfumed with synthetic TAGs together with extract-purified long OAcs.

**Figure supplement 6**. *D. arizonae* courtship preferences towards virgin females perfumed with different combinations of synthetic TAG together with extract-purified long OAcs.

---

females. Female remating was observed starting only at 8 hr post-mating (***Figure 5—figure supplement 2***). Taken together, these results show that the presence of male-transferred lipids on female cuticles is accompanied by a concomitant decrease in female attractiveness.

## Sex-specific lipids are anti-aphrodisiacs

The reluctance of males to court mated females could be the result of female rejection behavior or the presence of additional transferred compounds. To determine whether male-transferred lipids account solely for the loss of female attractiveness, we tested the preference of males from *D. arizonae*, *D. mojavensis*, and *D. navojoa* when presented with a choice between virgin females perfumed with evaporated solvent or extracts from ejaculatory bulbs. In these species, fewer males chose to initiate courtship with extract-perfumed females (***Figure 5E***). *D. arizonae* was also significantly less likely to copulate with bulb-perfumed females, whereas *D. mojavensis* showed a tendency to avoid perfumed females (***Figure 5—figure supplement 3***). *D. navojoa* only rarely copulated under these experimental conditions, likely because males prefer to court in the presence of more flies. However, the few males that copulated preferred to do so with control females. These results suggest that the use of these sex-specific TAGs as anti-aphrodisiacs is conserved across several species.

Further analysis of *D. arizonae* indicated that the higher the concentration of the extract, the greater the aversion exhibited by males. A minimum dose of extract from approximately 0.5 ejaculatory bulbs was needed to achieve a significant behavioral effect (***Figure 5E***). Copulation choice was similarly affected (***Figure 5—figure supplement 3***). Ejaculatory bulb extract from immature males

(and therefore, containing negligible amounts of TAGs and long OAcs) did not suppress male courtship (*Figure 5E*; *Figure 5—figure supplement 3*). Thus, courtship inhibition is not a generalized aversion to other compounds extracted from ejaculatory bulb tissue. Since virgin females were used, female rejection behavior was not a major contributing factor to copulation choice. Furthermore, it is unlikely that female rejection behavior was triggered by females' sensory feedback from perfuming. *D. arizonae* males continued to avoid mating with perfumed females from which the major olfactory and gustatory organs have been removed (*Figure 5—figure supplement 4*). Taken together, the presence of male-specific long OAcs and TAGs on females was sufficient to fully recapitulate the loss of attractiveness observed in recently mated females.

## Sex-specific TAGs function synergistically with other lipids and exhibit stereospecificity

To determine whether TAGs and long OAcs function synergistically with each other, we examined male response to females perfumed with these two classes of compounds separately and together (*Figure 3—figure supplement 1* shows TLC separation). Perfuming with both long OAcs and TAGs strongly suppressed courtship initiation and copulation (*Figure 5F,G*). The presence of TAGs alone significantly reduced the likelihood of copulation (*Figure 5G*). Interestingly, long OAcs had an attractive effect when used alone (*Figure 5F*). These results suggest that both TAGs and long OAcs are needed to suppress courtship initiation, whereas TAGs are important for discouraging later stages of courtship.

We next tested whether individual TAG species plays a role in suppressing courtship. We synthesized four of the postulated TAGs along with a TAG containing stearic acid as racemic mixtures (*Table 1*). Additionally, individual (*R*)- and (*S*)-enantiomers were synthesized for a TAG species containing oleic acid, the most abundant of the sex-specific TAGs (*Mori, 2012*). Males were given a choice of solvent-perfumed females or females perfumed with a single TAG species together with long OAcs. Under these conditions, TAGs containing oleic acid or linoleic acid significantly suppressed courtship initiation (*Figure 5—figure supplement 5*). However, only the former was capable of suppressing copulation as well and at a dose of 75 ng per female (*Figure 5H*). Moreover, only the (*R*) configuration of this TAG was bioactive (*Figure 5H*; *Figure 5—figure supplement 5*). Perfuming females with the (*S*)-enantiomer showed no significant effect on male choice, signifying that courtship aversion is specific to the stereochemistry of the TAG and not due to general avoidance of a foreign molecule or masking of female aphrodisiacs.

To examine the possibility that TAGs function synergistically with each other, we paired several combinations of synthetic TAGs together with long OAcs. At a dose of 37.5 ng per fly, none of the TAGs were effective by themselves. However, two combinations of TAGs reduced copulation: 1-steroyl-2,3-ditigloyl glycerol together with either 1-oleoyl-2,3-ditigloyl glycerol or 1-linoleyl-2,3-ditiglyoyl glycerol (*Figure 5I*). Both combinations also inhibited courtship initiation (*Figure 5—figure supplement 6*). Additionally, courtship initiation was affected by the combination of long OAcs with 1-palmitoleyl-2,3-ditiglyol and 1-oleyl-2,3-ditiglyol (*Figure 5—figure supplement 6*). Thus, low amounts of several TAG species, inactive on their own, can act in synergy with each other to suppress male courtship

**Table 1.** Synthetic TAGs used in this study

| Calculated [M + K]$^+$ | Fatty acyl components*,† | Long chain fatty acid |
| --- | --- | --- |
| 531.31 | C16:1 (n-7)-C5:1-C5:1 | Palmitoleic acid |
| 533.32 | C16:0-C5:1-C5:1 | Palmitic acid |
| 557.32 | C18:2 (n-6)-C5:1-C5:1 | Linoleic acid |
| 559.34 | C18:1 (n-9)-C5:1-C5:1 | Oleic acid |
| 561.36 | C18:0-C5:1-C5:1 | Stearic acid |
| 559.34 | C18:1 (n-9)-C5:1-C5:1 (*R* isomer) | Oleic acid |
| 559.34 | C18:1 (n-9)-C5:1-C5:1 (*S* isomer) | Oleic acid |

*Synthesized as racemic mixtures unless otherwise noted.
†Notation indicates number of carbons followed by number of double bonds for each fatty acyl component; double bond position indicated in brackets.

and copulation. It is notable that none of the TAG combinations tested were effective without long OAcs despite our finding that a mixture of TAGs purified from extract was by itself sufficient to deter male attraction. It may be the case that a combination of several different TAG species is needed for courtship inhibition without the presence of long OAcs.

## Discussion

Lipid and protein compounds transferred from male to female *Drosophila* during copulation are known to inhibit courtship from other males, trigger rejection behaviors in mated females, and serve as nutrition to aid in fertilization and oogenesis in the female (*Zawistowski and Richmond, 1986*; *Bownes and Partridge, 1987*; *Wigby et al., 2009*; *Yew et al., 2009*). Our results show that TAGs are a novel class of mating-related pheromones that are used by males to manipulate the post-mating attractiveness of females. It remains to be determined whether sex-specific TAGs or their hydrolyzed side chains serve other functions such as nuptial gifts (*Gwynne, 2008*) or defensive compounds (*Will et al., 2000*). Earlier studies of *D. arizonae* and *D. mojavensis* cuticular lipids using GCMS found mostly linear and branched long-chain hydrocarbons, including dienes, trienes, and methyl-branched alkenes, 29–38 carbons in length (*Toolson et al., 1990*; *Etges and Jackson, 2001*). TAGs were not previously observed likely due to analytical limitations; under standard GCMS conditions, heavier molecules such as TAGs are not detected. The presence of TAGs in cuticular extracts is usually attributed to fat leaking from internal stores due to cuticular rupture. However, several indicators make it clear that sex-specific TAGs are exogenously secreted. First, signals for specialized TAGs were detected only from the ejaculatory bulb housed in the anogenital region of males but not virgin females, indicating a discrete, sexually dimorphic site of secretion. Second, sex-specific TAGs are structurally distinct from conventional TAGs found in other *Drosophila* tissues (*Carvalho et al., 2012*; *Guan et al., 2013*). Last, short UV-LDI MS analysis times (below 30 s, corresponding to 1200 laser shots) causes minimal perturbation of the cuticle; breakdown of the outer layers appear only after 2000 laser shots (*Yew et al., 2011*).

### Synergy between pheromones

Synergistic interactions between multiple sensory cues in chemical communication have been described in several forms, in many cases involving a combination of enantiomers (*Borden et al., 1976*) or a blend of molecules from the same chemical class (*Lecomte et al., 1998*; *Srinivasan et al., 2008*). Famously, the honey bee queen uses a blend of least nine different fatty acids and alcohols secreted by multiple glands (*Keeling et al., 2003*) The parasitic wasp *Lariophagus distinguendus* was recently reported to use TAGs together with a branched alkane to promote mating behavior (*Kühbandner et al., 2012*). Food odors are also known to synergize with aggregation pheromones in beetles (*Lin et al., 1992*) and with sex pheromones in *Drosophila* (*Grosjean et al., 2011*). Pheromonal synergism between completely different classes of molecules is rare and may be a mechanism to increase combinatorial complexity. Interestingly, only some of the *D. arizonae* TAGs appear to play a role as an anti-aphrodisiac. The quiescent stereoisomers could be used as potential future chemical cues (*Niehuis et al., 2013*).

### Relationship of sex-specific TAGs to diet and environment

Sex-specific TAGs found on desert-adapted drosophilids are the first examples of natural products bearing combinations of branched and linear fatty acyl side chains. Conventional naturally occurring TAGs found in plant oils and animal fat typically consist of linear fatty acyl moieties that have 16, 18, or 20 carbons (*Nelson et al., 2008*). Although medium and short chain fatty acyls have been found in TAGs from, respectively, whale blubber (*Litchfield et al., 1971*) and bovine milk fat (*Breckenridge and Kuksis, 1968*), they are only in combination with linear and even-numbered carbon acyls. In contrast, sex-specific TAGs from flies exhibit an unusual combination of short and long chain acyl components with odd and even numbers of carbons. To what extent can these unconventional structures be attributed to diet? Previous and current work has shown that altering the diet of desert-adapted flies results in a quantitative change in hydrocarbons and sex-specific triglycerides (*Figure 4*) (*Toolson et al., 1990*; *Etges et al., 2006*; *Etges et al., 2009*; *Yew et al., 2011*). Notably, many of the substrates on which the drosophilids subsist and oviposit contain compounds that can be toxic for other animals. For example, desert-adapted *D. arizonae* and *D. mojavensis* feed on fermenting cacti, which have high levels of triterpene glycosides, medium chain fatty acids, and sterol diols, compounds which can serve

as toxic plant defense chemicals (*Fogleman and Danielson, 2001*). Similarly, *Drosophila subquinaria* and *Drosophila recens* are found exclusively on mushrooms that are rich in secondary metabolites such as isoprenoids and fatty acids in a variety of lengths, from 4 to 26 carbons (*Wandati et al., 2013*; *Pedneault et al., 2008*) but also contain high levels of toxic compounds like alpha-amanitin (*Jaenike et al., 1983*; *Courtney et al., 1990*). Slime fluxes, on which *D. virilis* and *D. robusta* can be found, have large bacterial communities that can be inhospitable to other drosophilids (*Carson and Stalker, 1951*; *Powell, 1997*). Bacterial wetwood infections have been shown to produce acetate, butyrate, valerate, hexanoate, and propionate (*Ward and Zeikus, 1980*), which could be directly incorporated into the TAGs or serve as precursors for branched or linear fatty acids, as has been observed for nitidulid beetles (*Carpophilus* spp) (*Bartelt and Weisleder, 1996*). The ability of drosophilids that produce sex-specific TAGs to thrive on these specialized substrates alludes to the possibility that enzymes used for detoxification may have been adapted for TAG synthesis. Notably, cytochrome P450 monooxygenases have been identified in other insects as playing a crucial role in detoxification and cuticular lipid synthesis (*Li et al., 2004*; *Qiu et al., 2012*). In bark beetles, it has been suggested that a key cytochrome P450 enzyme used in pheromone synthesis was previously used for detoxification (*Blomquist et al., 2010*). Alternatively, conservation of TAG expression may be more related to phylogenetic effects than functional adaptation (*Oliveira et al., 2011*).

A second pathway for TAG synthesis relies on de novo production of precursors. Each of the desert-adapted drosophilids tested in this study are still able to produce TAGs despite being raised on standard laboratory fly media. Thus, desert-adapted drosophilids are capable of using precursors from the environment and synthesizing the components de novo though significant quantitative differences are found for some TAG molecules. Based on these observations, we expect that lab-raised flies are likely to have quantitative differences in lipid profiles compared to natural populations because of the differences between the natural diet and a highly simplified lab diet. Both pathways for precursor synthesis are used by numerous Coleoptera beetle species for the production of aggregation pheromones (*Tillman et al., 1999*). The ability to utilize different production pathways may enable insects to switch host plants while preserving conspecific signaling. Elucidation of the biochemical pathways underlying TAG synthesis is needed to better understand whether dietary sequestration or de novo production is preferentially used by desert-adapted drosophlids and to determine the ancestral state of TAG production.

In this study, we have identified an unusual chemical class of pheromones in the form of triacylglycerides and described their function as anti-aphrodisiacs. Specialized TAGs were prevalent across other *Drosophila* species and may also be found in other insect orders and have other functions. For example, triolein in the fire ant, *Solenopsis invicta*, acts as a brood pheromone and has application as bait when combined with toxicants (*Bigley and Vinson, 1975*). The chiral 1,2-dioleyl-3-palmitolyl glycerol is also a brood pheromone in the honey bee, *Apis mellifera* (*Koeniger and Veith, 1984*). Taken together, triacylglycerides represent a broadly conserved and largely overlooked class of pheromones. Ultimately, to understand the evolutionary origin of these unusual molecules, it will be important to determine the behavioral function of sex-specific TAGs in other species and the underlying biosynthetic pathways. Using a broad range of analytical methods for chemical profiling will expand the detectable range of chemical classes used in communication. Correlating gene expression in the ejaculatory bulb across multiple TAG-expressing species will enable us to identify candidate biosynthetic enzymes and to provide molecular markers that will allow the evolution and function of this surprising chemical phenotype to be traced.

## Materials and methods

### Fly stocks and husbandry

*D. arizonae*, *D. aldrichi*, and *D. navojoa* in this study were wild caught from Las Bocas, Sonora, Mexico, in March 2009. *D. mojavensis* were caught from Santa Catalina Island, California and *D. wheeleri* from Punta Onah, Sonora, Mexico, in November 2007 by sweep netting over fermenting bananas. All stocks are available from WJE at the University of Arkansas. *Drosophila melanica* and *D. robusta* were obtained from UC San Diego *Drosophila* Stock Center. *D. virilis*, *D. hydei*, *D.americana*, *D. buzzatii*, and *D. montana* were obtained from Ehime-Fly *Drosophila* Stocks of Ehime University. Flies were reared on autoclaved yeast-sucrose-cornmeal-agar food or food supplemented with added banana (ca. 110 g/20 half-pint bottles) and cactus powder (ca. 2.3 g/20 half-pint bottles; Nopal cactus powder,

Oro Verde, Mexico) in a 23.3°C room on a 12:12 hr light:dark cycle at 60% humidity level. Adult flies were transferred to fresh food contained in half pint bottles every 3–5 days for female oviposition. After pupal eclosion, all emerging adults were sexed under $CO_2$ every 5 hr during the day. Virgin females were grouped in groups of 20–30 individuals in a new food vial, whereas males were isolated individually to keep them socially naive. The flies were allowed to reach sexual maturity (8–10 day old) at 23.3°C before behavioral analysis.

## Cuticular lipid extraction and thin layer chromatography purification

Ejaculatory bulbs from ca. 500 mature males were dissected and soaked in hexane in a 1.8 ml glass vial with a Teflon-lined cap (Wheaton, Millville, New Jersey, USA) for 20 min. The extract was placed in a clean glass vial, evaporated with $N_2$, and stored at −20°C until analysis. To obtain individual fractions, extract was overlaid onto a 10 × 10 cm thin layer chromatography silica plate (Merck, Darmstadt, Germany) and developed with a solution of hexane/diethyl ether/acetic acid (90:9:1; per vol). Silica from fractions containing the male-specific TAGs and acetates were scraped into a disposable borosilicate glass Pasteur pipette (15 cm length) stuffed with glass wool fiber (Pall Corporation, Ann Arbor, Michigan, USA) and eluted with hexane. The contents were divided into 3 aliquots, evaporated under a gentle stream of $N_2$, and kept at −20°C until analysis.

## Direct tissue thin layer chromatography of individual ejaculatory bulbs

Individual ejaculatory bulbs were dissected from 9-day-old *D. arizonae* males raised on either standard yeast-sucrose-cornmeal-agar food for two generations or cactus and banana-supplemented fly food. Each bulb was placed in its own lane on a 10 × 10 cm thin layer chromatography silica plate (Merck, Darmstadt, Germany). To release the contents and ensure complete elution, each bulb was first gently punctured then overlaid 10 times with 0.5 µl of hexane, allowing the solvent to fully evaporate between each solvent application. The plate was run in a solution of hexane/diethyl ether/acetic acid (90:9:1; per vol), developed with primuline (0.1% in 20% acetone), and imaged with the Gel Doc XR system (Bio-Rad Laboratories, Inc., USA) using Quantity One software (v 4.5.2, Bio-Rad Laboratories, Inc., USA). Intensities of the bands in the image were processed and analyzed using ImageJ (v 1.43, NIH, USA) to produce a plot of peaks according to brightness of the bands. Intensity values of the fractions were normalized to the intensity of the control band at the origin.

## Transesterification of TAGs and short chain FA standards

200 µl of methanolic HCl (Supelco Analytical, Sigma–Aldrich Co., St. Louis, MO) was added to dried, crude whole fly extract from about 500 flies and incubated for 1.5 to 2 hr at 60°C with occasional vortexing. After the acid-based catalysis, the reaction was cooled on ice, followed by the addition of 50 µl of water and 50 µl of hexane, and brief vortexing. The hexane layer (which contains the fatty acid methyl esters) was removed for GCMS analysis. Concurrently, synthetic standards containing 5 carbons (tiglic acid, trans-2-pentenoic acid, trans-3-pentenoic acid, and 3-methyl crotonic acid [TCI Chemicals Co., Tokyo, Japan]) were treated with the same reactions. Methyl angelate (TCI Chemicals Co.) was not treated.

## Chemical derivatization of long OAc fraction

The long OAc fraction obtained from TLC was derivatized with 200 µl of 0.2M KOH in 80% isopropanol for 2 hr at 60°C. After incubation, 50 µl of 1M HCl was added and evaporated under a gentle stream of $N_2$. 200 µl of hexane was added prior to analysis by DART MS and GCMS.

### Synthetic sex-specific TAGs

Synthesized TAGs used in this study are shown in *Table 1*. Synthesis procedures were previously described (*Mori, 2012*).

### Ultraviolet-laser desorption ionization mass spectrometry (UV-LDI MS)

UV-LDI MS analysis and the procedures for preparing the flies were described in detail previously (*Yew et al., 2009, 2011*). Measurements were performed on a QStar Elite (ABSciex) orthogonal time-of-flight mass spectrometer equipped with an intermediate pressure oMALDI2 source and a $N_2$ laser ($\lambda$ = 337 nm, 40 Hz repetition rate, 200 µm beam diameter, pulse duration 3 ns). Ions are generated in a buffer gas environment using 2 mbar of $N_2$. Individual flies were attached to a cover slip with adhesive tape and mounted onto a custom-built sample plate. During data acquisition, the anogenital region was irradiated for 30 s, corresponding to 1200 laser shots. Mass accuracy for the mass spectrometer

was approximately 20 ppm. Elemental composition and number of double bonds are predicted from exact mass measurements.

## Low energy collision-induced dissociation (CID) tandem MS

CID spectra were acquired on a linear ion trap mass spectrometer (Thermo Fisher Scientific LTQ, San Jose, CA) that has been modified to allow ozone-induced dissociation (OzID) experiments (*Thomas et al., 2008*; *Brown et al., 2011*). Methanolic solutions of lipid samples (ca. 10 µM) in the presence of either sodium or lithium acetate (ca. 10 mM) were infused into the electrospray ionization source of instrument with a flow rate of 5 µl/min; a spray voltage of 5 kV; a capillary voltage of 21 V; a tube lens voltage 125 V; and the temperature of the heated transfer-capillary was set to 275°C. To acquire CID spectra of triacylglycerol alkali metal adduct ions, the $[M + Na]^+$ or $[M + Li]^+$ were isolated with an isolation width of 1–2 Da and a normalized collision energy of 32–35% was applied.

## Ozone-induced dissociation MS

Individual TAG species generated by electrospray ionization of the extract was mass-selected within an ion trap mass spectrometer where they were exposed to ozone vapor. The resulting gas-phase ion–molecule reaction facilitates targeted oxidative dissociation of carbon–carbon double bonds present in the acyl chains. Fragmentation of the ozonide leads to formation of characteristic aldehyhde and Criegee ions with a mass indicative of the positions of each double bond. To determine the carbon–carbon double bond positions within TAGs, alkali metal adduct ions were mass-selected within the modified linear ion trap mass spectrometer (see above) and allowed to react with ozone seeded in the helium buffer gas (*Thomas et al., 2008*). To acquire OzID spectra, ions were isolated in the absence of collision energy and the reaction time (set by adjusting the activation time parameter within the XCalibr instrument control software) was typically 5–10 s per scan. OzID spectra reported here correspond to the average of at least 50 scans. Reaction of ozone with carbon–carbon double bonds in the TAG acyl chains produces fragment ions that identify their position within the chain. Location of each double bond is indicated by the traditional nomenclature '*n-x*' where '*n*' refers to the number of carbon atoms in the chain and subtracting '*x*' provides the location of the double bond (i.e., *x* represents the position of the double bond with respect to the methyl terminus).

## Gas chromatography mass spectrometry (GCMS)

Prepared extracts were re-dissolved in 60 µl of hexane and transferred into GCMS vials (Supelco). Analysis was run in a 5% phenyl-methylpolysiloxane (DB-5, 30 m length, 0.32 i.d., 0.25-µm film thickness, Agilent) column and GCMS QP2010 system (Shimadzu) with an initial column temperature of 50°C for 2 min and increment to 300°C at a rate of 15°C/min in splitless mode.

## Analysis of TLC fractions and dissected tissues by fly-assisted laser desorption ionization mass spectrometry (FALDI MS)

Fly wings were detached and washed in a solution of chloroform/methanol (2:1, vol/vol) to ensure existing cuticular hydrocarbons were completely removed. The wings were then attached onto a MS customized sample plate with double-sided tape. The fraction-containing vials were re-dissolved with 20 µl of hexane each and 2 µl from each vial was pipetted onto different wings. The sample plate was placed into the UV-LDI MS instrument and the fractions analyzed using the same parameters for UV-LDI MS analyses of intact flies. For more details see *Yew et al., 2011*.

## Direct analysis in real time mass spectrometry (DART MS)

Analysis was performed using the following ion source settings: the gas heater was set to 200°C; the glow discharge needle was set at 3.5 kV. Electrode 1 was set to +150 V and electrode 2 was set to +250 V. Helium gas flow was set to 2.5 L/min. Under these conditions, TAGs were detected as $[M + NH_4]^+$ molecules and long OAcs were detected as $[M + H]^+$ molecules. Clean borosilicate glass capillaries (World Precision Instruments) were used for sampling the solution. The capillary was placed in the DART stream for 5 s. Polyethylene glycol (Sigma–Aldrich) was used as calibrant. Analysis was done with MassCenter (version 1.3.0.1) (JEOL) program.

## Courtship assays

To generate mated female targets in assays where males choose virgin or recently mated females, naive males were first paired with virgin females in 35 × 10 mm tissue culture dishes (Nunclon, Denmark)

and observed for copulation. Immediately after copulation, the dish containing the copulated pair was placed on ice to anaesthetize the flies temporarily. In clean tissue culture dishes, a mated female and a virgin female are placed into each dish and allowed to recover for 45 min to an hour before performing the courtship assay. New socially naive males are aspirated into each dish and assayed for first courtship event lasting more than 5 s, including wing vibration, foreleg tapping, proboscis extension, and copulation. At least 20 trials of each set of experiments were performed.

For perfuming assays, virgin females were perfumed by lightly vortexing eight individuals in a 1.8-ml glass vial with Teflon-lined caps containing extract or evaporated solvent (control) for 3 bouts of 20 s each with 5–10 s rest between each bout. A fly from each vial was checked for perfuming efficiency using UV-LDI MS, whereas the other 7 were used for the courtship assay. Approximately, 20% of contents from the extract were transferred to all eight flies during perfuming (*Billeter et al., 2009*). Therefore, each fly was perfumed with the equivalent of 2.5% of the total concentration from the vial.

In assays using synthetic TAGs (*Table 1*), an amount of 750 ng or 75 ng per fly was used. Where TAGs are paired, each female was perfumed with 37.5 ng of each TAG for a total 75 ng per fly. *Figure 5J* shows the spectra of perfumed females, indicating that the amounts perfumed are moderate compared to mated females.

## Phylogenetic analysis

The phylogram was generated using Mesquite 2.75 (*Maddison and Maddison, 2011*; http://mesquiteproject.org). Distances and primary diets were based on the previous studies (*Throckmorton, 1975*; *Lemeunier et al., 1986*; *Spicer, 1991*; *Jeffs et al., 1994*; *Russo et al., 1995*; *Powell, 1997*; *Shoemaker et al., 1999*; *Remsen and O'Grady, 2002*; *Gao et al., 2007*; *Reed et al., 2007*; *Routtu, 2007*; *Flores et al., 2008*; *Mcdermott and Kliman, 2008*; *Oliveira et al., 2012*; *Curtis et al., 2013*).

## Acknowledgements

We thank YN Chiang, JY Chua, SH Ng, WC Ng, A Pirkl, S Shankar, and KJ Tan for technical support and suggestions. M Ritchie, S Wong, K Dreisewerd, H Luftmann, and G Pohlentz provided preliminary measurements and helpful discussion.

## Additional information

### Funding

| Funder | Grant reference number | Author |
|---|---|---|
| Australian Research Council | DP0986628, DP120102922, CE0561607 | Stephen J Blanksby |
| Singapore National Research Foundation | NRF-RF2010-06 | Joanne Y Yew |
| Alexander von Humboldt Foundation | | Joanne Y Yew |
| National Science Foundation | EF-0723930 | William J Etges |

The funders had no role in study design, data collection and interpretation, or the decision to submit the work for publication.

### Author contributions

JSRC, SRE, HTP, SJB, JYY, Conception and design, Acquisition of data, Analysis and interpretation of data, Drafting or revising the article; KM, Conception and design, Drafting or revising the article, Contributed unpublished essential data or reagents; QLK, Conception and design, Acquisition of data, Analysis and interpretation of data; WJE, Analysis and interpretation of data, Drafting or revising the article, Contributed unpublished essential data or reagents

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
