## [Decision Letter]

Thank you for sending your work entitled “Sex-specific triacylglycerides are widely conserved in *Drosophila* and mediate mating behavior” for consideration at *eLife*. Your article has been favorably evaluated by a Senior editor, a Reviewing editor, and 2 reviewers, both of whom, Wittko Franke and Leslie Vosshall, have agreed to reveal their identity.

The Reviewing editor and the reviewers discussed their comments before we reached this decision, and the Reviewing editor has assembled the following comments to help you prepare a revised submission.

All reviewers agreed that this well written manuscript described a significant expansion of our knowledge of pheromonal signaling in Drosophila. In brief the paper describes the use of sophisticated mass spectrometry techniques to characterize a small family of triacylglycerides (TAGs) that play a role in pheromonal communication in several species of Drosophila. In addition to defining the biological roles of this recently discovered family, the manuscript more generally illustrated the potential for new analytical chemical techniques to reveal previously overlooked biological phenomena.

This general enthusiasm for the manuscript was muted by some issues that are described below.

1) Both naturally occurring TAGs and synthetic TAGs are used in this study, and the failure to distinguish what has been definitively established about the structures of the two needs to be addressed. The synthetic TAGs, which are based on a 2012 paper with the Mori laboratory, have well defined structures. The naturally occurring TAGs are the issue, and the concerns are best illustrated by Figure 3—figure supplement 3 and Figure 3. The issue with the supplemental figure is that the structures shown on the right are not unequivocally established by the MS data shown on the left. For example, the first MS (A) shows that the structure is a triacylated glycerol with acyl groups containing 2, 5 and 18 carbon atoms, but it does not establish that they are connected to the glycerol in the order shown. In addition the ozonolysis experiments show that the C18 acyl group has a double bond at the position shown, but it does not establish whether it is *cis* or *trans*. Thus the drawing implies a degree of structural specificity that is not established in this or previous papers. The authors make assumptions, admittedly plausible assumptions, about unknown structural features. The best way forward would be more careful characterization of purified materials. At a minimum, the substitution pattern of the glycerol should be established, and the *cis*-*trans* stereochemistry could be left ambiguous – wavy lines – in the structural drawings. The substitution pattern could be established through synthesis as isolating enough material would likely prove difficult.

Figure 3 establishes that synthetic TAGs and their mixtures can have similar biological effects to the natural TAGs, but that in itself does not establish the structures of the natural TAGs. The pheromone literature has many examples where compounds similar but not identical to the natural ones can have very similar biological effects.

Finally, the reviewers suggested that as sophisticated chemical analysis and synthesis play such an important role in this manuscript, the chemistry should be moved from the supplementary material into the main body of the paper. An important feature of *eLife* is the ability to have as many figures and as much space as necessary to tell the story.

2) The authors assume that male rejections of TAG-perfumed virgins is not due to female rejection behavior, but they did not explicitly test this. An alternative in which female chemosensory detection of TAGs on their own cuticle could switch them into a post-mating state. This admittedly less likely alternative explanation could be tested quickly.

3) The authors mention that the TAGs might have a dietary origin. While it might be challenging to test with wild flies, it should be possible to vary the laboratory diet, especially plus/minus cactus powder to note any effect on TAG amount or composition. Absent that kind of evidence, it would be best to curtail speculation about the source of TAG precursors.

---

## [Author Response]

*1) Both naturally occurring TAGs and synthetic TAGs are used in this study, and the failure to distinguish what has been definitively established about the structures of the two needs to be addressed. The synthetic TAGs, which are based on a 2012 paper with the Mori laboratory, have well defined structures. The naturally occurring TAGs are the issue, and the concerns are best illustrated by*
Figure 3—figure supplement 3
*and*
Figure 3*. The issue with the supplemental figure is that the structures shown on the right are not unequivocally established by the MS data shown on the left. For example, the first MS (A) shows that the structure is a triacylated glycerol with acyl groups containing 2, 5 and 18 carbon atoms, but it does not establish that they are connected to the glycerol in the order shown. In addition the ozonolysis experiments show that the C18 acyl group has a double bond at the position shown, but it does not establish whether it is* cis *or* trans*. Thus the drawing implies a degree of structural specificity that is not established in this or previous papers. The authors make assumptions, admittedly plausible assumptions, about unknown structural features. The best way forward would be more careful characterization of purified materials. At a minimum, the substitution pattern of the glycerol should be established, and the* cis*-*trans *stereochemistry could be left ambiguous – wavy lines – in the structural drawings. The substitution pattern could be established through synthesis as isolating enough material would likely prove difficult*.

Figure 3
*establishes that synthetic TAGs and their mixtures can have similar biological effects to the natural TAGs, but that in itself does not establish the structures of the natural TAGs. The pheromone literature has many examples where compounds similar but not identical to the natural ones can have very similar biological effects*.

We have established the substitution pattern for 8 of the TAGs based on the relative abundances of product ions in the CID spectra (Figure 3; Figure 3—figure supplement 4). It is well established that upon dissociation, loss of substituents at *sn-1* and *sn-3* of the glycerol backbone will be more abundant than at *sn-2* ([25] and [26]). Hence, based on the relative intensities of peaks in the CID spectrum, it is possible to deduce backbone position. Using this principle, we were able to assign the long chain FA component to *sn-1* or *3* for 8 of the TAGs. The assignments for 4 of the TAGs are supported by CID analysis of the synthetic versions that produced nearly identical dissociation spectra (Figure 3—figure supplement 6, Figure 3—figure supplement 7 and Figure 3—figure supplement 8).

With respect to *cis-trans* stereochemistry of double bonds, we are able to deduce the geometry of one of the TAG molecules. [48] observed that *cis* vs *trans* double bonds differed in terms of overall abundance and relative abundances of OzID product ions since trans-alkenes react faster with ozone. OzID analysis of the natural TAG and a synthetic version (synthesized with *cis* double bond geometry) produced similar spectra in terms of fragment ion abundances, supporting the *cis* assignment. In all other cases, we have placed a wavy line to illustrate ambiguity of the double bond configuration. We agree that synthesis of TAGs with variant substitution positions will be highly valuable for structural elucidation. The Mori laboratory is currently in the process of synthesizing TAGs with different substitution patterns of the glycerol, a procedure that is likely to require between 2-3 months. Once completed, we intend to perform the CID analysis and publish the results in a separate paper. However, for the current paper, we believe that the substitution patterns for the major TAG species (particularly ones that appear to have functional significance) that we have established should be sufficient.

*Finally, the reviewers suggested that as sophisticated chemical analysis and synthesis play such an important role in this manuscript, the chemistry should be moved from the supplementary material into the main body of the paper. An important feature of eLife is the ability to have as many figures and as much space as necessary to tell the story*.

We thank the reviewers for their enthusiasm regarding the chemistry. We have included a much more detailed schematic of the product ions result from CID MS and OzID analysis for the TAG species highlighted in Figure 3. These schematics highlight the chemical analysis and are intended to illustrate spectral interpretation. However, we have elected to keep other supporting CID and OzID spectra in supplemental figures because the interpretation of the spectra is already summarized in Figure 3.

*2) The authors assume that male rejections of TAG-perfumed virgins is not due to female rejection behavior, but they did not explicitly test this. An alternative in which female chemosensory detection of TAGs on their own cuticle could switch them into a post-mating state. This admittedly less likely alternative explanation could be tested quickly*.

We have addressed this intriguing possibility by testing females from which different sensory organs have been surgically removed (antennae, maxillary palps, and first segment of the foreleg; Figure 5—figure supplement 6). In each case, males still preferred to court with unperfumed females indicating that females ‘ability to detect TAGs on their own cuticle does not trigger rejection behavior or contribute in a significant way to influencing male avoidance.

*3) The authors mention that the TAGs might have a dietary origin. While it might be challenging to test with wild flies, it should be possible to vary the laboratory diet, especially plus/minus cactus powder to note any effect on TAG amount or composition. Absent that kind of evidence, it would be best to curtail speculation about the source of TAG precursors*.

We quantified levels of sex-specific TAGs from individuals raised on either standard fly media or media supplemented with cactus powder and banana. Thin-layer chromatography was performed on dissected ejaculatory bulbs from 10 flies for each condition. TAGs were quantified according to the intensity of the primuline-stained TLC band. The results indicate a significant reduction in several, but not all, of the sex-specific TAGs when flies are raised on standard media. We conclude that a laboratory diet has an effect on TAG production and likely contributes precursors or makes precursors more readily available for incorporation. We have added these data to the manuscript (Figure 4).

To further address this issue, we make reference in the Discussion to a previous publication that examined the effects of diet on overall cuticular lipid changes and found significant quantitative changes.